

# Acute warming tolerance ($CT_{max}$) in zebrafish (*Danio rerio*) appears unaffected by changes in water salinity

Eirik R. Åsheim[1,2,3], Anna H. Andreassen[3], Rachael Morgan[3,4], Mireia Silvestre[3] and Fredrik Jutfelt[3,5]

[1] Institute of Biotechnology, Helsinki Institute of Life Science (HiLIFE), University of Helsinki, Helsinki, Finland
[2] Organismal and Evolutionary Biology Research Programme, Faculty of Biological and Environmental Sciences, University of Helsinki, Helsinki, Finland
[3] Department of Biology, Norwegian University of Science and Technology, Trondheim, Norway
[4] Department of Biological Sciences, University of Bergen, Bergen, Norway
[5] Department of Biological and Environmental Sciences, Faculty of Science, University of Gothenburg, Gothenburg, Sweden

## ABSTRACT

Tolerance against acute warming is an essential trait that can determine how organisms cope during heat waves, yet the mechanisms underlying it remain elusive. Water salinity has previously been suggested to modulate warming tolerance in fish and may therefore provide clues towards these limiting mechanisms. Here, using the critical thermal maximum ($CT_{max}$) test, we investigated whether short (2 hours) and long (10 days) term exposure to different water salinities (2 hours: 0–5 ppt, 10 days: 0–3 ppt) affected acute warming tolerance in zebrafish ($N = 263$). We found that water salinity did not affect the warming tolerance of zebrafish at either time point, indicating that salinity does not affect the mechanism limiting acute warming tolerance in zebrafish at these salinity ranges, and that natural fluctuations in salinity levels might not have a large impact on acute warming tolerance in wild zebrafish.

## INTRODUCTION

With climate warming, average water temperatures are increasing, and in many places, temperature extremes are increasing in frequency, duration and maximum temperatures (*Seneviratne et al., 2014*). In some aquatic habitats, heat waves may push populations to or above their acute warming tolerance limits. Such situations may already be occurring in some coral reefs (*Genin et al., 2020*), and freshwater systems (*Morgan et al., 2019*; *Wegner et al., 2008*). As the increasingly severe heat waves will increase the impacts of extreme temperatures, there is increased research interest in understanding the mechanisms behind warming tolerance. Portions of this publication were previously published as part of a preprint (*Åsheim et al., 2022*).

Corresponding author
Eirik R. Åsheim,
eirik.asheim@helsinki.fi

The most commonly used measure of acute warming limits in aquatic ectotherms is the critical thermal maximum test ($CT_{max}$), where the water temperature is gradually increased until the animal displays a loss of balance or locomotor function (*Becker & Genoway, 1979*; *Lutterschmidt & Hutchison, 1997*; *Morgan, Finnøen & Jutfelt, 2018*; *Morgan et al., 2020*; *Desforges et al., 2023*). The temperature at loss of equilibrium (LOE) is commonly used as the endpoint for actively swimming fish. However, the mechanisms causing the LOE are partly unknown (*Andreassen et al., 2022*; *Ern, Andreassen & Jutfelt, 2023*). With a lack of understanding of the underlying physiology, it is also difficult to predict how temperature may interact with other factors (*Ern, Andreassen & Jutfelt, 2023*; *Jutfelt et al., 2024*).

One such other factor is water salinity. Salinity stress can occur in aquatic environments through processes such as migration between water bodies, evaporation, ice melt, flooding and precipitation (*Kültz, 2015*). As climate change affects weather patterns, many fish populations such as those residing in estuaries, rivers and lakes will likely experience changing water salinities (reviewed by *Gillanders et al., 2011*; *Mosley, 2015*; *Robins et al., 2016*; *van Vliet et al., 2023*).

Examining how water salinity affects warming tolerance can be useful for predicting interactions of factors during heat waves in nature. If water salinity modulates warming tolerance limits, then differences in precipitation or evaporation may affect the warming tolerance of some populations. Additionally, modulation of warming tolerance by other factors may give information on which physiological mechanisms are controlling these traits. There have been a few indications in the recent literature suggesting that water salinity can modulate warming tolerance in both freshwater and seawater fishes, but there has been no consistent direction of effects with salinity manipulation (Fig. 1). Some studies report a reduction in warming tolerance with increased salinity (*Morgan et al., 2019*; *Shaughnessy & McCormick, 2018*), some show increased warming tolerance with increasing salinity (*King & Sardella, 2017*; *Metzger, Healy & Schulte, 2016*), some show a peak in warming tolerance around some optimal salinity (*Haney & Walsh, 2003*; *Sardella, Sanmarti & Kültz, 2008*; *Loeb & Wasteneys, 1912*), whilst others report no effect (*Davis et al., 2019*; *Hines et al., 2019*) (Table S1). Several different mechanisms linking osmotic stress and acute warming tolerance have been suggested to be responsible where $CT_{max}$ has been affected, such as oxygen availability (*via* changes in blood haematocrit and lactate) (*Shaughnessy & McCormick, 2018*), expression of heat shock proteins (*Metzger, Healy & Schulte, 2016*), and changes in gill permeability causing an osmoregulatory compromise (*Haney & Walsh, 2003*; *Wood & Eom, 2021*). However, none of these suggested mechanisms have so far been confirmed.

A field study by *Morgan et al. (2019)* suggested a reduction in $CT_{max}$ with higher salinities. In that study, $CT_{max}$ of wild zebrafish (*Danio rerio*) correlated negatively with water salinity, with fish from higher salinity streams (measured as conductivity) showing lower warming tolerance. It was, however, not clear if that effect was due to water salinity or some other confounding factor of the sampled river systems. This curious observation that stream salinity appeared to affect warming tolerance, together with the extensive amount of research tools available for the species, suggested that zebrafish could make a good study system for research on the effects of salinity on warming tolerance.

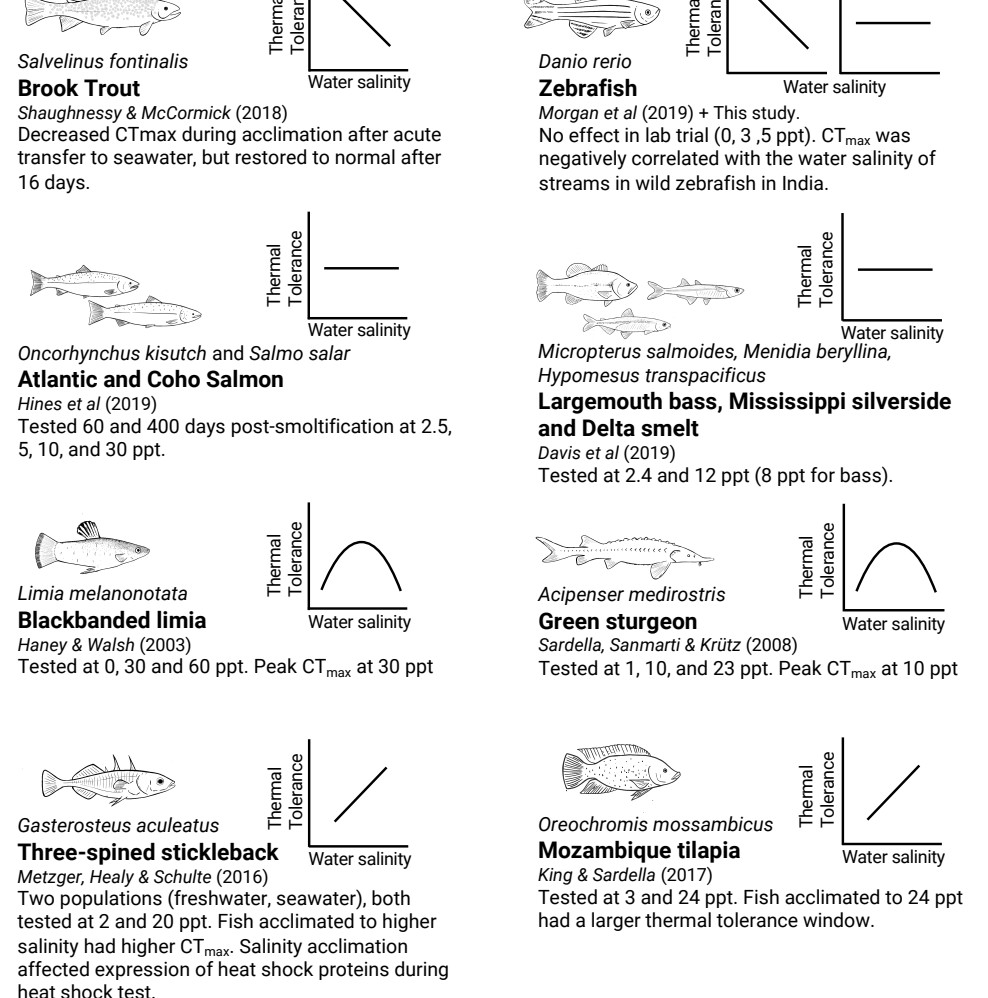

**Figure 1** Summary of currently available reports from studies testing $CT_{max}$ under various water salinity treatments in different fish species. The small graphs indicate the general, simplified pattern of the change in $CT_{max}$ as salinity increases (increased, decreased, no change, or peaked). Different species of fish and different experimental settings give different effects of salinity on thermal tolerance. Note: *Shaughnessy & McCormick (2018)*; *Morgan et al. (2019)*; *Hines et al. (2019)*; *Davis et al. (2019)*; *Haney & Walsh (2003)*; *Sardella, Sanmarti & Kültz (2008)*; *Metzger, Healy & Schulte (2016)*; *King & Sardella (2017)*.

To test the findings by *Morgan et al. (2019)*, to explore the zebrafish as a model species for interactions between salinity and warming tolerance, and to build on the existing body of knowledge on thermal tolerance, we designed an experiment testing $CT_{max}$ of zebrafish after exposure to different salinity treatments. We were also interested in knowing whether a potential effect is immediate (*e.g.*, due to acutely disturbed plasma ion concentrations), and if such an effect is modulated by longer exposures and acclimation to the water salinity levels. Therefore, we tested fish at two different time points: after two hours and after ten days of salinity exposure.

## MATERIALS & METHODS

### Animal research permit

This experiment was approved by the Norwegian Animal Research Authority (Norwegian Food Safety Authority, permit number: 8578). Relevant local guidelines and policies were followed in all experimental procedures and animal care. We conducted this experiment at the animal research facilities of the Natural Sciences Building of the Norwegian University of Science and Technology, Trondheim, Norway.

### Fish rearing

Fish used in this study were randomly selected from the first filial generation of wild-caught zebrafish from West Bengal, India (*Morgan et al., 2019*; *Morgan et al., 2022*). These fish were about 1 year and 4 months old at the start of the experiment. For the duration of the experiment, fish were held in 45 L glass aquaria ($50 \times 30 \times 30$ cm) filled to 35 L, at 26 °C, and under a 12/12 day/night cycle. In each tank, an ornamental aquarium plastic plant was used to enrich the tank environment, and water was filtrated and aerated using two bacteria-seeded air-lift sponge filters. Fish were fed ad-libitum twice a day using TetraPro energy flakes (Tetra®, Blacksburg, VA, USA). Tanks were inspected twice every day (once on Saturdays and Sundays) for signs of distress or to remove any dead fish. Criteria for euthanasia were established so that any fish showing acute distress (erratic swimming, loss of equilibrium) or prolonged distress (lying still, infections, wounds) were to be euthanized with an MS222 overdose (Tricaine Methanesulfonate, 0.6 g $l^{-1}$, sodium bicarbonate-buffered at 0.8 g $l^{-1}$), although no fish were identified to be in distress so that this was put into action. All fish were euthanized after the conclusion of the experiment.

### Salinity exposures and experimental design

The experimental design included four treatment groups of different salinity, and each treatment had two duplicate tanks. Prior to the experiment, the fish were kept at 0.5 ppt, which is considered optimal holding conditions for laboratory zebrafish (*Lawrence, 2007*). One group was kept at this concentration as a control group. A group kept at 1 ppt served as a "high" treatment, and a group kept at 5 ppt served as a "very-high" treatment. To see if low-salinity stress could also affect warming tolerance, we included a 0 ppt group using Millipore-filtered water. All fish were acutely exposed to the salinity treatments. Salinity was changed by moving the fish to a temporary tank while replacing the water in the experimental tank with water at the target salinity. The fish were kept for three days to habituate to their experimental tanks at 0.5 ppt before the salinity treatments started.

Each of the eight experimental tanks had at least 30 individuals in each tank. The total number of fish was 278, with 70 fish at 0 ppt, 69 at 0.5 ppt, 68 at 1 ppt, and 71 at 5 ppt. When water was exchanged, different levels of water salinity were achieved by adding natural sea salt (GC Rieber salt A/S, 98.0% NaCl) to aerated and carbon-filtered tap water and measured to 0.1 ppt accuracy with a digital water quality meter (YSI ProDSS, YSI Inc., Yellow Springs, OH, USA) using a conductivity measurement probe (ProDSS ODO/CT, YSI Inc, Yellow Springs, OH, USA). Tank water salinities were tested daily to check for drift due to evaporation and feeding. A small increase in salinity over the experiment was

**Table 1  Model summary.** Statistical summary for the mixed-effect model on $CT_{max}$. F and $p$ values were calculated using Satterthwaite's method with type III tests. The mixed model was fitted using restricted maximum likelihood. $CT_{max}$ was modelled to be linearly affected by salinity (Salinity, 0-5 ppt), time of exposure ($Time_{long}$, 2 h or 10 days, coded 0 or 1), as well as a term including the specific time taken from start of exposure to the first test, accounting for variation in time until the first test ($Time_{short}$, expressed in minutes)(Table S2). $Time_{short}$ was set to zero for all fish tested after 10 days (*i.e.*, only included for those tested after approx 2 h). Interactions were included between salinity and both time factors. No significant effect was found for any of the included factors.

| Fixed effects | Estimate | 2.5% CI | 97.5% CI | Sum Sq | Mean Sq | DenDF | F (type II) | $p$ |
|---|---|---|---|---|---|---|---|---|
| Intercept | 40.713 | 40.494 | 40.931 | | | | | |
| Salinity (ppt) | 0.066 | −0.138 | 0.272 | 0.017 | 0.017 | 28.912 | 0.099 | 0.557 |
| $Time_{short}$ (min) | 0.000 | −0.001 | 0.001 | 0.007 | 0.007 | 28.689 | 0.041 | 0.841 |
| $Time_{long}$ (10 days) | −1.972 | −20.69 | 16.659 | 0.006 | 0.006 | 28.694 | 0.038 | 0.848 |
| Salinity:$Time_{short}$ | 0.000 | −0.002 | 0.001 | 0.016 | 0.016 | 28.889 | 0.096 | 0.758 |
| Salinity:$Time_{long}$ | 3.143 | −15.512 | 21.889 | 0.016 | 0.016 | 28.912 | 0.095 | 0.760 |

| Random effects | Std.dev |
|---|---|
| $CT_{max}$ test group | 0.133 |
| Tank and time | 0.000 |
| Residual | 0.412 |

proportional to the salinity of the tanks; the two tanks with the lowest salinity increased by 0.02 and 0.03 ppt, and the two tanks with the highest salinity increased by 0.10 and 0.13 ppt.

Zebrafish is a freshwater species, and the observations available on the salinities they experience in the wild indicate that they live at low salinities around $0.1-0.6$ ppt (*Morgan et al., 2019*; *Spence et al., 2006*; *Sundin et al., 2019*). However, a laboratory experiment testing acute salinity tolerance in zebrafish found no effect on metabolism or nitrogen excretion at salinities as high as 20 ppt (*Uliano et al., 2010*). Knowing this, we chose 5 ppt as the highest salinity treatment for our experiment as we wanted a strong positive control in case we found no effect, while staying far below what we knew these fish could survive acutely. The fish were regularly inspected after the initial change of salinity (twice per day, and once per day on Saturdays and Sundays), looking for any change in behaviour or activity that could indicate distress. After two days, three fish displayed signs of inactivity (*e.g.*, lying still on the bottom) in one of the 5 ppt tanks, which indicated that the 5 ppt salinity treatment was too high. Therefore, the salinity concentration was decreased to 3 ppt for the remaining of the ten-day treatment. This is still 6 times higher than the recommended salinity for zebrafish (*Lawrence, 2007*), and 50 times higher than the $0.06 \pm 0.02$ ppt ($\pm$SD) salinity measured in their native water bodies in India (*Morgan et al., 2019*; *Sundin et al., 2019*).

### Testing of acute warming tolerance ($CT_{max}$)

The fish were tested for acute warming tolerance at two different time points. Half of each treatment group (each group $n = 30-39$) were tested for $CT_{max}$ 130–150 min after the start of salinity exposure (Table S2), and the other half after ten days of salinity exposure. The fish tested after 130–150 min were tested for $CT_{max}$ at salinities 0, 0.5, 1, and 5 ppt, while those tested after ten days were tested for $CT_{max}$ at salinities 0, 0.5, 1 and 3 ppt.

The $CT_{max}$ tests were conducted as by *Morgan, Finnøen & Jutfelt (2018)*. Briefly, the $CT_{max}$ method consists of a plexiglass tank ($33 \times 23 \times 24$ cm) with two compartments

separated by a mesh: one heating compartment and one fish arena. The heating compartment contained a 300 W heater coil placed within a custom-made heating chamber and attached to a water pump to maintain a homogenous temperature within the tank (*Morgan, Finnøen & Jutfelt, 2018*). The tank was filled with 9.9 L of 26 °C water at the corresponding salinity concentration for each treatment. A group of 7–11 fish were placed in the arena simultaneously and the temperature was gradually increased by a rate of 0.3 °C per minute. The temperature at which the fish lost equilibrium (defined as disorganised swimming for three seconds, usually rolling during swimming) was recorded with 0.1 °C precision using a high-precision thermometer (Testo-112, Testo, Titisee-Neustadt, Germany). Other behavioural abnormalities were noted but not quantified with precision. At the point of LOE, the fish were removed from the fish arena and returned to control temperature and salinity where they rapidly recovered. All fish were fasted for 24 h before the $CT_{max}$ test and there was 100% survival after the tests.

## Statistical analysis

For statistical analysis, each individual counted as one observation belonging to one of the five salinity treatments (coded 0, 0.5, 1, 3, 5 ppt), one of the time points (coded 0 for 2 h and 1 for 10 days), one tank (coded 1 or 2), and one $CT_{max}$ test tank (coded 1 or 2). Additionally, for fish in the 2-hour treatment, we included a term to account for the precise time between the start of the salinity treatment to the $CT_{max}$ test (coded in the number of minutes), to account for some variation in this. This term was set to zero for fish in the 10-day treatment. To test for an effect of salinity and time-in-treatment on $CT_{max}$, we fitted a linear mixed effect model using $CT_{max}$ (°C) as a response variable. Explanatory variables were salinity treatment, time-point, and time-in-treatment. Interaction terms were included between salinity and time-point, and salinity and time-in-treatment. Finally, two random effects on the intercept were included: one for each combination of tank and time-point, and one for each group of fish tested for $CT_{max}$ together. F and *p* values were calculated using Satterthwaite's method with type III tests. The mixed model was fitted using restricted maximum likelihood. All analysis was performed using *R* v4.2.1 (*RCore Team, 2021*) *via Rstudio* v2022.7.0.548 (*RStudio Team, 2022*). R packages used were *lmerTest* v3.1.3 (*Kuznetsova, Brockhoff & Christensen, 2017*) for mixed model analysis; *ggplot* v3.3.6 (*Wickham, 2016*) for plotting; and the *tidyverse* v1.3.1 (*Wickham et al., 2019*) package collection for general data management.

## RESULTS

$CT_{max}$ was not significantly affected by water salinity or treatment time (Table 1, Fig. 2). The mean $CT_{max}$ for all individuals was 40.8 ± 0.03 °C (Mean ± SE, $N = 263$). Three fish were showing signs of inactivity after two days in the 5 ppt treatment (lying still on the bottom of the tank), resulting in us lowering the salinity of the highest treatment to 3 ppt for the remainder of the experiment (as described in 'Methods'). In the highest salinity treatment (3–5 ppt), five (of 71) individuals died during the experiment. These deaths happened on days three ($n = 1$), seven ($n = 1$) and ten ($n = 3$). Furthermore, six individuals in the highest salinity treatment were removed from the $CT_{max}$ test before reaching $CT_{max}$ because they

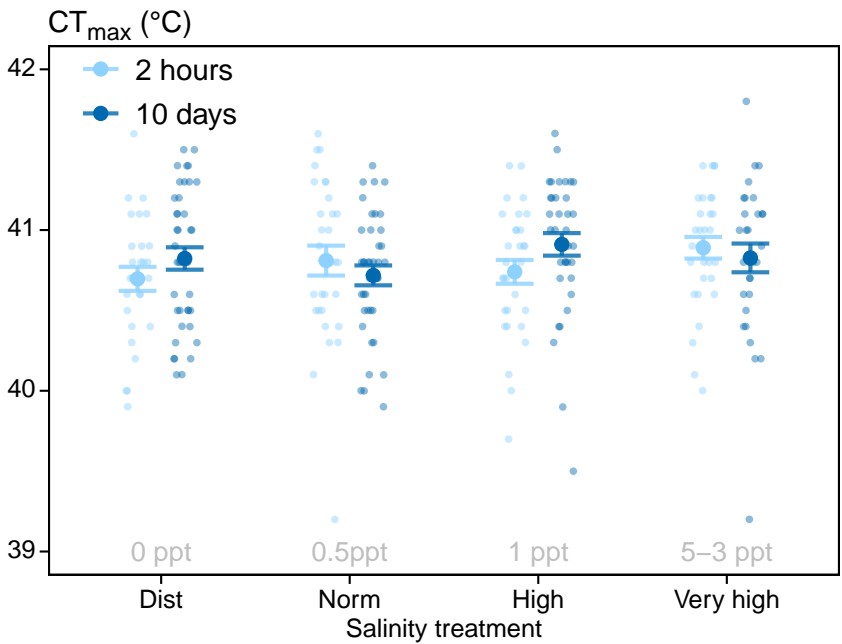

**Figure 2 CT$_{max}$ results.** CT$_{max}$ for all individuals ($N = 263$) tested at four different salinity treatments and two treatment durations (2-hour and after 10-day exposure). The 3 ppt salinity was only tested for the 10-day group, and the 5 ppt salinity was only tested for the 2-hour group. Mean group results and SE are indicated by large points and error bars. Small points indicate CT$_{max}$ of single individuals. Points are horizontally jittered for readability.

displayed abnormal behaviour during heating (see discussion). Four individuals jumped out of the CT$_{max}$ test tank and were excluded from the analysis (three from the 0 ppt treatment and one from the 1 ppt treatment. Total number of fish included in the analysis was 263.

## DISCUSSION

In this experiment, we aimed to test a potential connection between water salinity and warming tolerance as reported in wild zebrafish by *Morgan et al. (2019)*, and to better understand interacting factors on acute warming tolerance. We found no effect on CT$_{max}$ from a 2-hour exposure to salinities ranging from 0-5 ppt, or a 10-day exposure to salinities ranging from 0-3 ppt (Table 1, Fig. 2).

The highest salinities used (3 and 5 ppt) were 6 and 10 times higher than normal holding salinities for zebrafish (0.5 ppt), and 50 and 70 times higher than what has been observed in the wild for this species (0.06 ppt). While both salinities are comparably high, these salinities are still hypoosmotic to the fish's internal ion concentrations (approximately 9 ppt), and as such may not require large changes in the osmoregulatory activity of the fish, but only a modulation of their ion- and water intake (*Kültz, 2015*). Here, we chose relatively low salinities as there is little information on what salinities zebrafish can tolerate long-term. Our observation of three fish lying immobile in one of the 5 ppt tanks after

two days indicates that this salinity is too high for long-term survival, but this may also be due to the acute transfer to this salinity. It is possible that the fish could have endured those salinities if more gradually exposed to them. Further studies on interactions between thermal- and osmotic stress in zebrafish would benefit from more research on the ranges of survivable salinities for these fish in the long term, as well as blood measurements to indicate the degree of osmotic stress experienced. Nonetheless, our results show that high, but still hypoosmotic water salinities do not affect acute warming tolerance in zebrafish.

During $CT_{max}$ testing of fish from the highest salinity treatment (at ten days), an unexpected behaviour occurred in which fish would sometimes perform a sudden burst in swimming, followed by a cramped seizure-like spasm lasting a few seconds that caused loss of equilibrium. This happened at temperatures down to 10 °C lower than the normal $CT_{max}$ and was first mistaken as their final LOE at $CT_{max}$, hence the premature removal of six fish before their true $CT_{max}$. This happened to several more individuals and we found that they always recovered after these cramping events and would continue swimming until reaching the normal LOE reaction at similar temperatures to other individuals. The total number of individuals this happened to was not recorded, but it only occurred in the highest salinity treatment after two weeks of exposure and has not been observed in other experiments. These seizure-like events could indicate that the highest salinity treatment had some negative effect on the tested fish and that there could be some connection between acute warming tolerance and water salinity. As these observations were anecdotal, follow-up studies are needed to confirm and quantify this effect.

Acute warming tolerance was previously reported to be negatively correlated with habitat water salinity in a survey of wild zebrafish populations (*Morgan et al., 2019*; *Sundin et al., 2019*). That result does not match the observed robustness of $CT_{max}$ to even larger salinity differences in the current study. We believe the present result is more reliable due to the higher sample size and broader range of salinities. The discrepancy may be due to an additional factor(s), associated with stream salinity that confounded the correlation observed in *Morgan et al. (2019)*, or other factors differing between the wild- and laboratory setting (*Morgan et al., 2022*). In any case, these results show that variation in salinity beyond what was measured in natural streams by *Morgan et al. (2019)* does not seem to influence warming tolerance in zebrafish. However, more extreme variations in salinity may occur in freshwater environments due to various events such as heatwaves, droughts, and saltwater intrusion, all of which are increasing in frequency with climate change (reviewed by *Mosley, 2015*; *van Vliet et al., 2023*). The excact magnitude of salinity variation during such events varies with a range of environmental factors, but two studies have reported median increases around 21% (*Jones & Van Vliet, 2018*) and 24% (*Graham, Bierkens & Van Vliet, 2024*). In our study, we used far greater increases in water salinity and observed no effect on acute warming tolerance, indicating that acute warming tolerance might not be affected by salinity increases during droughts and heatwaves in zebrafish at large. However, our results are likely only applicable for warming challenges that can be resolved in the short term (hours); whether changes in salinity may affect thermal tolerance- and/or performance over longer durations (days) is an open question.

To get a better overview of the diverse relationship between salinity and $CT_{max}$ in previous studies, we compiled a list of published papers where water salinity was manipulated experimentally, and where $CT_{max}$ was tested as the outcome variable (Fig. 1. Table S1). Compared to these previous studies, our study adds to the list of studies showing no effect, together with *Davis et al. (2019)* and *Hines et al. (2019)*. A notable difference between the studies showing an effect and those not showing one is the salinity tolerance of the species used. This study and the study by *Hines et al. (2019)* used relatively low water salinities (up to 5 ppt and 12 ppt), while studies reporting an effect (*Haney & Walsh, 2003*; *King & Sardella, 2017*; *Metzger, Healy & Schulte, 2016*; *Sardella, Sanmarti & Kültz, 2008*; *Shaughnessy & McCormick, 2018*) used far higher ranges (up to 25 ppt, and up to 60 ppt in one study). The studies showing an effect all used highly euryhaline species and acclimation times of at least one week, indicating that the effects on warming tolerance may be connected to salinity acclimation itself or through some other general stress response. Still, the study by *Hines et al. (2019)* found no effect despite using salinities up to 30 ppt and acclimation times up to 600 days. Overall, looking at the published literature on the subject, it looks like the association between salinity and warming tolerance is highly complex and species- and context-dependent.

## CONCLUSIONS

We explored whether the acute warming tolerance of zebrafish is affected by water salinity. We found no difference in $CT_{max}$ between fish tested at water salinities ranging from 0 to 5 ppt in the short term (2 h) or from 0 to 3 ppt in the long term (10 days). These results indicate that there is little interaction between acute warming tolerance and environmental salinity at these salinity ranges in zebrafish, and that variation in salinity levels encountered by wild zebrafish might not impact their acute warming tolerance.

## ACKNOWLEDGEMENTS

We acknowledge and thank Miriam Dørum and Jan Arvid Sand for their help with feeding and maintenance of the experimental setup.

### Funding

This experiment was supported by the Research Council of Norway (Norges Forskningsråd 62942). There was no additional external funding received for this study. The funders had no role in study design, data collection and analysis, decision to publish, or preparation of the manuscript.

### Grant Disclosures

The following grant information was disclosed by the authors:
The Research Council of Norway (Norges Forskningsråd 62942).

## Competing Interests

The authors declare there are no competing interests.

## Author Contributions

- Eirik R. Åsheim conceived and designed the experiments, performed the experiments, analyzed the data, prepared figures and/or tables, authored or reviewed drafts of the article, and approved the final draft.
- Anna H. Andreassen conceived and designed the experiments, performed the experiments, authored or reviewed drafts of the article, and approved the final draft.
- Rachael Morgan conceived and designed the experiments, performed the experiments, authored or reviewed drafts of the article, and approved the final draft.
- Mireia Silvestre conceived and designed the experiments, performed the experiments, prepared figures and/or tables, authored or reviewed drafts of the article, and approved the final draft.
- Fredrik Jutfelt conceived and designed the experiments, authored or reviewed drafts of the article, and approved the final draft.

## Animal Ethics

The following information was supplied relating to ethical approvals (i.e., approving body and any reference numbers):

This experiment was approved by the Norwegian Animal Research Authority (Norwegian Food Safety Authority, permit number: 8578).

## Data Availability

The full dataset and R script are available at Zenodo:

Åsheim, E. R., Andreassen, A. H., Morgan, R., Silvestre, M., & Jutfelt, F. (2023). Data files and R scripts for manuscript: Acute warming tolerance (CTmax) in zebrafish (Danio rerio) appears unaffected by changes in water salinity (Version 4) [Data set]. Zenodo. https://doi.org/10.5281/zenodo.8435610.

## Supplemental Information

Supplemental information for this article can be found online at http://dx.doi.org/10.7717/peerj.17343#supplemental-information.

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
