# Peer review of "Acute warming tolerance (CTmax) in zebrafish (Danio rerio) appears unaffected by changes in water salinity"

_PeerJ, doi:10.7717/peerj.17343_

## Round 0.1 · original submission · Major Revisions

Dear Authors

The reviewers have commented on your manuscript. You can find attached reports. Based on the comments and suggestions of the expert reviewers, unfortunately, a major revision is needed for your article.

I would like to request you check and correct the manuscript step by step based on the reports.

Sincerely yours

Reviewer 1 ·

Basic reporting

All the references used in this article are appropriated, but it would be interesting to add a few more to support some statements, especially in the introduction section. The format of the article does not entirely respect the standard format accepted by the journal, as the results and discussion sections have been merged, whereas it is indicated in the template that these sections must be separated from each other. Moreover, even if they appeared in the supplemental table 2, the statistical results are not fully reported in the results section.

Experimental design

No comment

Validity of the findings

Although this study shows that salinity does not affect the thermal tolerance of fish, the results are clearly stated regarding measures of thermal tolerance and the study is replicable.
However, the change in salinity during the project for the highest salinity makes the replicability of this part of the project more complicated, and the interpretation of the results is also affected.
Furthermore, there is a lack of clear information on the behavioural observations made on fish that were exposed to the highest salinity, as no real recorded results are presented in the article. That is, a part of the data on which some conclusions are based are not provided. Thus, some conclusions are not directly supported by the results presented in the manuscript.

Additional comments

The authors show with this article that the acute thermal tolerance of zebrafish could not be directly affected by water salinity. For this purpose, the authors set up a simple but effective and appropriate protocol to directly measure the effects of different salinity levels on thermal tolerance, with different exposure times. However, the change in salinity level during the project for the highest salinity level makes it somewhat difficult to interpret the results obtained in this group, especially since behavioural observations made in these fish in particular were unfortunately not recorded. I think that parts of the discussion section might we reworded to be sure that the conclusions presented in the article are based only on clearly recorded results, and that a discussion of the limits of the study, with the changes in the level of salinity during the project, is added.

Major comments:

Results and Discussion
Lines 198 to 208: It is really interesting to see that the highest level of salinity led to this unexpected behaviour for some fish. However, as no results have been recorded, I would not have put so much emphasis on these observations. Then, you should keep these results as perspectives added in the conclusion for example, and perhaps add a brief description of these results in the supplemental data.

Lines 210 – 222: You indicate here that the highest level of salinity in your study was 3 ppt. However, it was not the case for the entire exposure time. I think that it might be interesting to also discuss the fact that, after 2 days, the highest level of salinity has been decrease from 5ppt to 3. Moreover, you should confirm in the methods if the fish were exposed to 3 or 5 ppt for the 2-hours exposure.

Minor comments:

Introduction
Lines 35 and 38: You should choose between “heat wave” and “heatwave” in terms of writing.
Line 52: That would be great to add a reference here to explain how salinity stress can occur in aquatic environments.
Line 62: You should also add some references here to explain how plasma osmolality and ion composition could be involved in the mechanisms controlling thermal tolerance.
Line 74: A period is missing at the end of the sentence.

Material and Methods
Line 112: Remove the extra “any”.
Line 123: Was the water quality monitored several times a day or continuously?
Line 154: As the exposure was from 130 to 150 minutes, did you take into account the exposure time in your analyses by looking only at the “2-hours exposure” group?
Line 159: You wrote the tank dimensions with “x” at line 106, then you should choose a single way to write dimensions.
Line 174: You should explain directly here why some individuals were removed from the analyses, as you wrote at the start of the manuscript that you had 269 fish for this project.

Conclusions
Line 250: You should remove the capital letter from "Zebrafish", since you have written "zebrafish" throughout the article.

Figure 1: You should increase the graphs size and reduce the fish size. It would be informative if the salinity values tested in the studies were presented on the x-axis so that the thermal tolerance depending on the salinity level could appear more clearly on each graph.

Figure 2: You should indicate in the figure caption that fish were exposed to 5 ppt for 2 hours or 5 ppt for 2 days and then 3 ppt for the rest of the ten-day exposure.

Supplementary table 2: Was the salinity level in your model a continuous variable or a fixed effect? Also, since the fish in the "2 hour" and "10 day" groups were not exposed to the same level of salinity, was this taken into account in your analyses?

·

Basic reporting

There was only one typo I saw on line 112 where the repeated word needs to be deleted.

I thought one idea to improve the basic reporting and article structure could be to add a table detailing the statistics output (f and p values, degrees of freedom etc.) from the linear mixed effect model, as currently the results do not actually have any mention of the statistical output, only one mention that there was no detectable effect on line 195-196.

Experimental design

no comment

Validity of the findings

One finding that is mentioned several times is about a "temporary seizure effect" that occurred in the highest salinity treatment. But this effect was basically not at all quantified (line 205-206: "The total number of individuals this happened to was not recorded"), so the reader is left guessing whether it could have happened to most of the fish or some of them or just a few. Because it was not analysed or even quantified, I do not think it should be emphasised as a valid result, as has been done on lines 29, 207, 213 and 255.

Additional comments

line 141: which study was this?

line 228: The discrepancy may also be due to the different contexts of the experiments, as conditions in small laboratory tanks each with one plastic ornamental plant will be very different to conditions for surveys of wild zebrafish populations, so just having zebrafish in tanks alone can be expected to cause all kinds of effects.

Reviewer 3 ·

Basic reporting

I offer my respects to the researchers who prepared this valuable article. Below are my suggestions for the article.

General information that should be written under the "Introduction" heading is written in the "material-method" section (For example, the sentence between lines 138 and 140). In addition, it is thought that the information that should be given in the results is written in the "material-method" section. Therefore, I suggest a general re-evaluation of the writing of the article. I suggest that the results written in the "material-method" section be transferred under the "results" heading, or it is suggested that researchers write only the experimental conditions they have set up according to the values they find best in their preliminary study. It is suggested that the article should be evaluated as "short communication".

It is thought that an experiment was designed as a preliminary study. Accordingly, it is recommended to write the title according to the content. Although the title was written directly on the result of the study, when the experimental template and the results obtained are evaluated, it is thought that the title suggested by the authors is not appropriate.

I suggest citation for sentences between lines 50 and 54.

Experimental design

I suggest specifying the size/weight/age of the fish in the material-method section, and writing the camera feature that monitors the fish in line 144.

I already mentioned in "Basic reporting" section.

Validity of the findings

It is a valuable study since it is a subject that researchers have focused on in recent years. The fact that this study was carried out on zebrafish, which is a model species, strengthens the widespread effect of the study.

---

## Round 0.2 · Minor Revisions

Dear Authors

The reviewers have commented on your manuscript. You can find the attached reports. The manuscript is almost ready for acceptance, however, a minor revision is needed for your article based on the comments and suggestions of the expert reviewers,

I would like to request last time you check and correct the manuscript step by step based on the reports.

Sincerely yours

Reviewer 1 ·

Basic reporting

The corrections made by the authors suit me.

Experimental design

No comment

Validity of the findings

The corrections made by the authors suit me.

Additional comments

The corrections made by the authors suit me.

·

Basic reporting

I could see just three small changes for the use of English language: at line 72 add a full stop, line 270 change to "We explored", and for Table 1 caption, change the last word to "factors".

Experimental design

No comment

Validity of the findings

No comment

Additional comments

line 52-54: the details of this sentence are a bit vague. Rather than just writing about "some fish populations" and "some form of osmotic stress", you could list actual examples of specific fish populations, and of specific osmotic stresses. And they could be examples where the fish were not just "likely to experience" the stress effect, but where they were experimentally determined to experience it.

line 63-67: I guess all of the studies cited here are about freshwater fish? Please clarify about that in the text.

Line 38-39: This sentence was unexpected. Maybe a place where it could be fitted in better is near the end of the introduction, e.g. one idea is that it could be added in the middle of line 87, and changed to something like: "Portions of the text in this paper describing our experiment were previously published as part of a preprint (Åsheim et al., 2022). Overall, we were interested to know whether an interaction effect occurs and whether it may be immediate".

line 269: the introduction has information about more broad ecological considerations, such as effects of heat waves, precipitation, evaporation etc., but in the discussion and conclusions the present study's results are not referred back to these initial considerations. I think some extra text is needed in the discussion and conclusions about how these results have increased our understanding of how zebra fish populations will (or will not) be expected to react to heat waves and changes in precipitation and evaporation. Some extra text like this could also be added to the end of the abstract.

Reviewer 3 ·

Basic reporting

When the whole of this study was evaluated, it was seen that basic evaluations were made. This is not enough for a research paper to publish.

Experimental design

The experimental design was primarily evaluated in terms of the fish species used. Although zebrafish is a model organism, it is not appropriate to carry out this study on wildly caught fish.

Validity of the findings

Since no standardized experimental design has been created, the reproducibility of the results is a topic that needs to be discussed.

---

## Round 0.3 · Minor Revisions

Dear Dr. Åsheim,

The reviewers have commented on your manuscript. You can find the attached reports. The manuscript is almost ready for acceptance, however, a minor revision is needed for your article based on the comments and suggestions of the expert reviewers,

I would like to request last time you check and correct the manuscript step by step based on the reports.

Sincerely yours

·

Basic reporting

Line 145: please remove first capitalisation. On line 263 also.

Figure 1: correct misspellings for Largemouth bass and Mississippi on 2nd graph down on right.

Experimental design

no comment

Validity of the findings

no comment

---

## Round 0.4 · accepted · Accept

Dear Dr. Åsheim,

I would like to thank you and your co-authors for making the corrections and changes requested by the reviewers. I read and checked carefully your valuable article and I am happy to inform you that your article has been accepted for publication in PeerJ.

Best regards